# Medical–Legal Entomology in Action: Evaluation of Insect-Based Post-Mortem Interval Estimation in South Korean Death Investigations

**DOI:** 10.3390/insects16020231

**Published:** 2025-02-19

**Authors:** In-Seong Baek, Hyeon-Seok Oh, Yi-Re Kim, Min-Gyu Kang, Jae-Bong Jung, Sang-Hyun Park

**Affiliations:** 1Department of Biomedical Sciences, Kosin University, Wachi-ro 194, Busan 49104, Republic of Korea; is6630@nie.re.kr (I.-S.B.); tjr3863@naver.com (H.-S.O.); 04174@naver.com (Y.-R.K.); alsrb0886@gmail.com (M.-G.K.); 2Invasive Alien Species Team, National Institute of Ecology, Seocheon-gun 33657, Republic of Korea; 3Metropolitan Police Agency, Busan 47545, Republic of Korea; applebox@naver.com

**Keywords:** medico-legal entomology, minimum post-mortem interval (PMI-min), insect development, forensic investigations, Busan, South Korea

## Abstract

This study investigates the use of forensic entomology for estimating the minimum post-mortem intervals in three death cases from Busan, South Korea. Traditional methods are limited to 48–72 h post-mortem, prompting the need for alternative approaches. The insects collected from the bodies, particularly *Lucilia sericata* (Meigen) and *Chrysomya megacephala* (Fabricius), were analyzed using morphological and DNA identification. Corrected death scene temperatures were calculated using thermo-hygrometer data and meteorological records and analyzed via quadratic regression and Support Vector Machine models. The post-mortem interval (PMI) estimates differed from the victim’s last known activity by 1–2 days, likely due to the pre-colonization interval and other factors like weather, oviposition timing, mixed fly populations, and maggot-generated heat. This study emphasizes the need for integrating advanced tools like machine learning to refine PMI estimation methods and account for complex variables.

## 1. Introduction

In a criminal investigation, the minimum post-mortem interval (PMI-min) is critical, as it helps identify the victim and the individual or individuals involved in their death [1]. Forensic pathologists use post-death conditions such as *livor*, *algor*, and *rigor mortis* to estimate the time of death. However, these techniques are reliable only for the initial 48–72 h post-mortem [2]. Given this limitation, new methods for determining the PMI-min have emerged, including assessing RNA degradation in tissues [3] and measuring various substances in the vitreous humor [4]. Despite their potential, many of these methods are not commonly employed because of their imprecision and delay in obtaining results [5]. Therefore, alternative studies or methods are needed to estimate the PMI-min. Forensic entomology is one approach that focuses on presenting credible scientific evidence using insect data to resolve the details surrounding the deaths of victims [6,7,8].

Medical–legal entomology, a vital area of forensic science, utilizes the succession and developmental stages of necrophagous insects to enhance the accuracy of criminal investigations [9]. Succession data indicate the range of the post-mortem interval (PMI), whereas development data estimate the PMI-min. Most recently, developmental data have been refined to define the PMI-min, which marks part of the total PMI-min from when insects first colonize the body to the discovery of the oldest immature insects [10]. Although there have been instances in South Korea where the PMI-min was estimated using necrophagous insects [11,12,13], few studies have rigorously examined the accuracy of PMI-min estimations based on insect evidence. Validating PMI-min estimation using insect evidence is a critical process, as it not only helps to identify potential errors in the methodology but also provides opportunities for refinement and improvement in the accuracy and reliability of these forensic techniques. The aim of this study was to evaluate the difference between the estimated PMI-min and the last recorded activity of the deceased, using entomological evidence from three cases in Busan, South Korea (Figure 1). This was to verify the accuracy of current PMI-min estimation methods using the Support Vector Machine (SVM) and quadratic regression models and then to determine the ambient crime scene temperature and identify potential areas for refinement in medico-legal entomology.

## 2. Materials and Methods

### 2.1. Collection of Insect Evidence

All forensic cases received in 2022 that were infested with insect specimens were transferred to the Department of Medical Biology, Kosin University, for investigation. Personal data (age, sex, location where the corpses were found, possible causes of death, etc.), if available, were recorded. Photographs of corpses and insect specimens were obtained. Approximately 20 insect specimens were collected using long forceps, placed in clean plastic boxes, and immediately transported to the Department of Medical Biology at Kosin University. In cases where a limited number of fly larvae were collected, the specimens were preserved to serve as representative samples for further analysis. The larvae were divided into two groups, each consisting of approximately ten specimens. To halt further development, all larvae were immediately immersed in hot water at a temperature of approximately 70 °C for 30 s, for preserving the morphological features of the maggots [14]. The first group was allocated for DNA barcoding to enable species identification, while the second group was preserved in 70% ethanol for entomological evidence in the forensic investigations. During the DNA barcoding, the most developed samples were prioritized for analysis to estimate the minimum post-mortem interval. Additionally, samples with diverse morphological characteristics, such as surface texture and color, were selected to represent a variety of species.

### 2.2. Molecular Identification of Insect Specimens

The anterior region of each specimen was used for DNA extraction using the YesGTM Cell Tissue kit (GenesGen, Busan, Republic of Korea). The COI was amplified and then made up to a final volume of 20 µL through the following PCR mixture: 10 µL of 2X TOP simple premix-HOT (Enzynomics, Daejeon, Republic of Korea), 0.4 µL of each primer as was performed by Park and Shin [15], 7.2 µL distilled water, and 2–3 mL of DNA extract. The designed primer pair is described in Table 1. The PCR products were amplified using a VeritiproTM Thermal Cycler (Applied Biosystems, Waltham, MA, USA). The amplification products were analyzed on 1.5% agarose gels. Sequencing reactions were conducted using BigDye(R) Terminator v3.1 Cycle Sequencing Kits (Applied Biosystems, MA, USA) with an ABI PRISM 3730XL Analyzer (96-capillary type). For identification using DNA databases, each sequence was matched using the Basic Local Alignment Search Tool (BLAST) of the National Center for Biotechnology Information (NCBI, Bethesda, MD, USA).

### 2.3. PMI-Min Estimation

The corrected death scene temperatures were calculated using an electronic thermo-hygrometer (RC-5+, Elitech Co., London, UK), which has an accuracy of ±0.5 °C for temperature and ±3% for humidity. These measurements were taken at the scene, supplemented by temperature data from the nearest Korea Meteorological Administration stations: Busanjin-gu (Station 938) for Case 1, Busannam-gu (Station 942) for Case 2, and Haeundae (Station 937) for Case 3.

To estimate the ambient temperatures at the crime scenes, both the quadratic regression model and the SVM model were applied, following the methodology outlined by Jeong et al. [16]. In line with their recommendations, a minimum of 2 days’ worth of temperature data, recorded at one-hour intervals, was utilized to enhance the estimation accuracy. A comparison of the results from both models revealed that the SVM model demonstrated greater accuracy due to its ability to integrate multiple environmental factors, including wind speed, humidity, and rainfall. Although the quadratic regression model provided reliable results, the SVM model produced lower values for the Mean Absolute Difference (MAD) and Root Mean Square Error (RMSE), indicating superior predictive accuracy, particularly in complex environmental conditions. This dual-model approach allowed for more accurate past-temperature estimations, enhancing the precision of PMI-min calculations in both open and closed crime-scene environments.

The collected insect evidence was analyzed to estimate the PMI-min using growth models and developmental data from established studies. Specifically, the growth model of *Lucilia sericata* (Meigen) described by Shin et al. [17] and the developmental data of *Chrysomya megacephala* (Fabricius) detailed by Zhang et al. [18] were used. These studies provided comprehensive growth data at constant temperatures that were used to estimate the PMI-min by comparing the developmental stages of the collected insect specimens to the known growth rates at the recorded temperatures using accumulated degree hours (ADH).

## 3. Results

### 3.1. Case Circumstances

#### 3.1.1. Case 1

On 24 May 2022, at 16:00, the body of a 61-year-old male was found in a room within a house situated in an urban area of Busanjin-gu, Busan. The body was clothed in upper and lower garments and was found on the bed. The windows and doors were open, and the heating and cooling systems did not function. The head and parts of the legs were exposed, and the upper limbs were positioned under the chest. The entire body was brown (Figure 2), indicating that it was in the decay stage of decomposition, with body fluids pooled beneath it. Maggots were observed on the head, and a significant number of maggots were present in the anal region of the body. Fly pupae were observed near the entrance of the room, with no recorded rainfall occurring outside during the relevant period. Based on the temperature data measured at the scene using an electronic thermo-hygrometer and data provided by the nearest Korea Meteorological Administration, the average corrected temperature was calculated to be 26.2 °C using both the quadratic regression model and SVM analysis.

#### 3.1.2. Case 2

On 12 August 2022, at 14:50, the body of an 84-year-old female was discovered lying on her side in the living room of a residence located in an urban area of Nam-gu, Busan. The entrances and windows were open. The body had no underwear. The face and the upper half of the body had turned black. The legs showed blue discoloration. The body was shaded. The heating and cooling systems did not function. The body was in the decay stage of decomposition, with pooled body fluid beneath it (Figure 3). Third-instar larvae were present in the facial and neck regions, with no recorded rainfall occurring outside during the relevant period. Based on the temperature data measured at the scene using an electronic thermo-hygrometer and data provided by the nearest Korea Meteorological Administration, the average corrected temperature was calculated to be 28.9 °C using both the quadratic regression model and SVM analysis.

#### 3.1.3. Case 3

On 30 September 2022, at 11:50, the body of a 45-year-old male was discovered with his head suspended by a rope affixed to chopsticks, which were secured to the windows and window frames in a residence located in an urban area of Suyeong-gu, Busan. Despite the windows and doors being closed, the ambient outdoor temperature was recorded to be 21.5 ± 3.6 °C and the indoor temperature was 24.5 ± 0.8 °C, indicating a minimal temperature difference. The body was clothed with a short-sleeved t-shirt and jeans. The body was shaded. The lower right ankle of the seated body was naked (Figure 4). The body had turned black, and maggots were observed on the face and mouth. A high abundance of maggots and puparia were discovered beneath the duvets, floors, and footrests, with no recorded rainfall occurring outside during the relevant period. Based on the temperature data measured at the scene using an electronic thermo-hygrometer and data provided by the nearest Korea Meteorological Administration, the average corrected temperature was calculated to be 21.6 °C using both the quadratic regression model and SVM analysis.

### 3.2. Entomological Evidence and PMI Estimation

#### 3.2.1. Case 1

Subsequent identification confirmed the presence of Diptera, specifically puparia of *L. sericata* (Figure 5). The PMI estimated from entomological evidence was calculated by considering the developmental stage of the *L. sericata* puparia. According to Shin et al. [17], reaching this developmental state takes approximately 10 days and 1 h at an average temperature of 26.2 °C. This was determined using both the quadratic regression model and the SVM model for estimating past average field temperature. From 21 May, 13:00 to 24 May, 18:00, the *L. sericata* transitioned from the post-feeding stage to puparium formation. Between 18 May, 14:00 and 21 May, 13:00, the larvae were in the third instar stage, transitioning to post-feeding. Between 16 May, 14:00 and 18 May, 14:00, the development progressed from the second to the third instar. From 15 May, 13:00 to 16 May, 14:00, it progressed from the first to the second instar, while between 14 May, 15:00 and 15 May, 15:00, the development went from egg to the first instar. Based on this data, the PMI-min was estimated to be 14 May 2022, at 15:00 (Figure 6). Through police investigation, the victim’s last known activity was confirmed via a pharmacy receipt timestamped on 13 May 2022, at 16h00.

#### 3.2.2. Case 2

Subsequent identification confirmed the presence of *L. sericata* and *C. megacephala* third-instar larvae (Figure 7). According to Shin et al. [17], the development time for *L. sericata* third-instar larvae is approximately 3 days and 11 h at an average temperature of 28.9 °C, which was calculated using the SVM model to estimate Past Average Field Temperature. In comparison, *C. megacephala* larvae develop faster, taking about 1 day and 22 h at the same temperature, as noted by Zhang et al. [18]. Because *L. sericata* remains on the body for a relatively long period, it was chosen for a more accurate PMI estimation as a first colonizer.

From 10 August, 21:50 to 12 August, 14:50, the *L. sericata* transitioned from the second to the third instar. Between 9 August, 23:50 and 10 August, 21:50, the larvae developed from the first to the second instar, and from 9 August, 03:50 to 9 August, 23:50, they progressed from eggs to the first instar. Based on this developmental timeline (Figure 8), the PMI-min was estimated to be 9 August 2022, at 03:50. Through police investigation, the victim’s last action was confirmed by a call on 7 August 2022.

#### 3.2.3. Case 3

Subsequent identification confirmed the presence of Diptera, third-instar larvae of *L. sericata* and *C. megacephala* (Figure 9). The PMI estimated from entomological evidence was calculated by mainly considering the development of the *L. sericata* to third-instar larvae, which, according to Shin et al. [17], takes approximately 14 days and 8 h at 21.6 °C, calculated by the SVM for estimating Past Average Field Temperature. In comparison, *C. megacephala* puparia develop faster, taking about 7 days and 17 h at the same temperature, as noted by Zhang et al. [18]. Because *L. sericata* remains on the body for a relatively long period, it was chosen for a more accurate PMI estimation as a first colonizer.

From 27 September, 01:50 to 30 September, 11:50, the *L. sericata* progressed from the post-feeding stage to the puparium stage. Between 22 September, 12:50 and 27 September, 01:50, the larvae transitioned from the third instar to the post-feeding stage. From 19 September, 09:50 to 22 September, 12:50, they developed from the second instar to the third instar. Between 17 September, 22:50 and 19 September, 09:50, the larvae progressed from the first instar to the second, and from 16 September, 11:50 to 17 September, 22:50, they developed from eggs to the first instar. Based on this developmental timeline (Figure 10), the PMI-min was estimated to be 16 September 2022, at 11:50. The medical examiner estimated the PMI-min to be 15 September 2022.

## 4. Discussion

When investigating a corpse, the PMI-min can vary from a few hours to days, months, or even years. Forensic pathologists commonly use post-mortem changes, such as *livor*, *algor*, and *rigor mortis*, to estimate the time of death. However, these methods are reliable only for the initial 48–72 h post-mortem [2]. As decomposition progresses over an extended period, estimating the PMI using these traditional methods becomes increasingly difficult and imprecise. This limitation necessitates the use of alternative approaches for scenarios with longer decomposition periods.

Forensic entomology offers a viable solution for estimating the PMI-min in such cases. By comparing the known developmental stages of insects on the corpse to how long laboratory flies of the same species take to reach that stage at a known temperature, forensic entomologists can estimate the PMI-min. This method utilizes the predictable developmental rates of necrophagous insects, primarily blowflies, which are among the first to colonize decomposing remains [8].

This study summarizes three forensic entomology cases investigated in Busan, South Korea, comparing each PMI-min estimated through insect evidence to the time of death estimated via final evidence and autopsy. To improve the accuracy of the PMI-min, an SVM model, developmental data, and quadratic regression were used. In all three cases, a total of two species, *L*. *sericata* and *C. megacephala*, were identified, with *L. sericata* being present in all three cases (Table 2). These two species are known to predominantly appear in urban areas within South Korea [11,19]. Consequently, the discrepancies between the estimated times in this study and those estimated from the final evidence and autopsy were 23 h to 2 days (Table 3). There are no statistical differences when comparing the PMI using entomological evidence with that determined by autopsy. The slight discrepancies may have arisen because of the differences in the pre-colonization interval (PCI). The PCI refers to the interval between death and the initial colonization of the corpse by insects. According to Matuszewski et al. [20], the minimum PCI value for adult *L. sericata* in an environment of 14.0–25.1 °C is reported to be 0.2 of a day. By applying the PCI value to Case 3, which corresponds to the specified temperature range, the PMI-min was calculated to be approximately 20 h.

The PCI is affected by several factors, like location of the corpse (e.g., indoor, outdoor), ambient temperature, and volatile organic compounds (VOCs) [21,22,23,24]. According to Reibe and Madea [25], insect colonization of corpses placed indoors was delayed up to 24 h compared with those placed outdoors. VOCs emitted from decomposing remains attract carrion insects, and their release is influenced by temperature [26,27,28,29,30]. Although the PCI is often overlooked, it is crucial to provide a more accurate PMI estimation. Studies from abroad, such as those by Faris et al. [31], have considered the PCI, including the exposure, detection, and acceptance phases. In South Korea, research has predominantly focused on the growth rates of necrophagous insects and the carrion insect fauna present on carcasses in different regions. However, there is a notable lack of studies that specifically evaluate or measure arthropod activity during the critical phases of decomposition. This gap in the research poses significant challenges for forensic entomologists in accurately predicting the PMI. Despite the availability of reliable development data and proper specimen collection methods, the absence of comprehensive studies addressing arthropod behavior during the decomposition phases makes precise PMI estimation difficult.

Several factors contribute to the uncertainty surrounding the PMI estimates. These include the assumptions outlined by Catts [32], such as insect activity during inclement weather (e.g., heavy rain and at night); the timing of blowfly oviposition; mixed fly populations and broods; seasonal effects on maggot-generated heat; the effect of maggot-generated heat itself; the impact of species arriving out of sequence; the availability of gravid fly recruits at the scene; and the effects of drugs, toxins, and parasitoids on maggot development. Studying these factors will enhance the accuracy of PMI estimations.

More accurate results can be obtained by incorporating the effects of these variables into PMI-min estimations. For instance, research on the environmental conditions that influence insect behavior and development can significantly refine PMI estimates. Additionally, not only the ecological study of necrophagous insects but also the integration of advanced technologies, such as machine learning models [15], to predict environmental effects on insect development could further improve accuracy.

Our findings indicate that the PMI-min estimates derived from the entomological evidence closely align with the last known activities of the deceased and the estimated times of death provided by the forensic pathologists. For instance, the estimated PMI-mins did not significantly differ from the last known activity times of the deceased or the times determined by autopsy. This concordance supports the notion that PMI estimation using insect evidence is reasonable and reliable. Although traditional methods primarily focus on insect development and real-time temperature data, PMI estimation remains valid and essential. Future development of integrated models that include other factors, such as the PCI, will enhance the accuracy of PMI estimations. Despite the current limitations, forensic entomology continues to be one of the most reliable methods for determining PMIs and plays a crucial role in forensic investigations.

Future research should prioritize the expanding datasets to encompass a broader range of decomposition stages and environmental conditions. Such an approach will contribute to refining forensic entomological methodologies and applications, thereby ensuring more accurate and reliable PMI estimations. By addressing these gaps and integrating comprehensive environmental data, forensic entomologists can significantly enhance the precision of PMI estimates, ultimately aiding in the resolution of criminal investigations.

## Figures and Tables

**Figure 1 insects-16-00231-f001:**
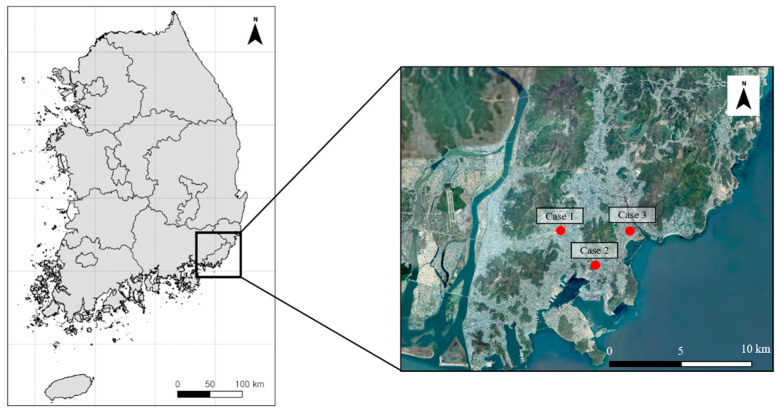
Map of Busan location in South Korea.

**Figure 2 insects-16-00231-f002:**
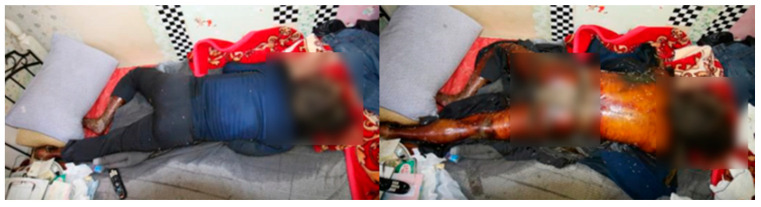
Photographic evidence of Case 1, a 61-year-old male, demonstrating the position in which the body was discovered, surroundings, concentration of insects, and physical state of decomposition.

**Figure 3 insects-16-00231-f003:**
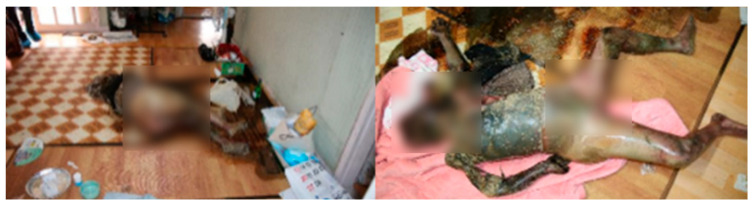
Photographic evidence of Case 2, an 84-year-old female, demonstrating the position in which the body was discovered, surroundings, concentration of insects, and physical state of decomposition.

**Figure 4 insects-16-00231-f004:**
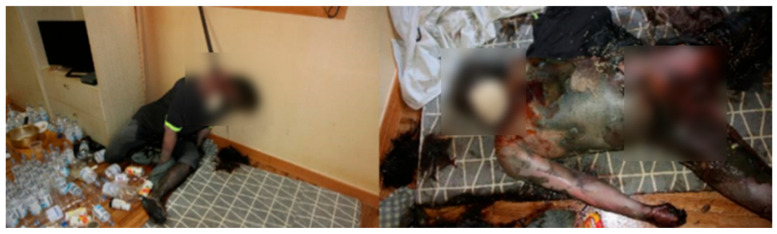
Photographic evidence of Case 3, a 45-year-old male, demonstrating the position in which the body was discovered, surroundings, concentration of insects, and physical state of decomposition.

**Figure 5 insects-16-00231-f005:**
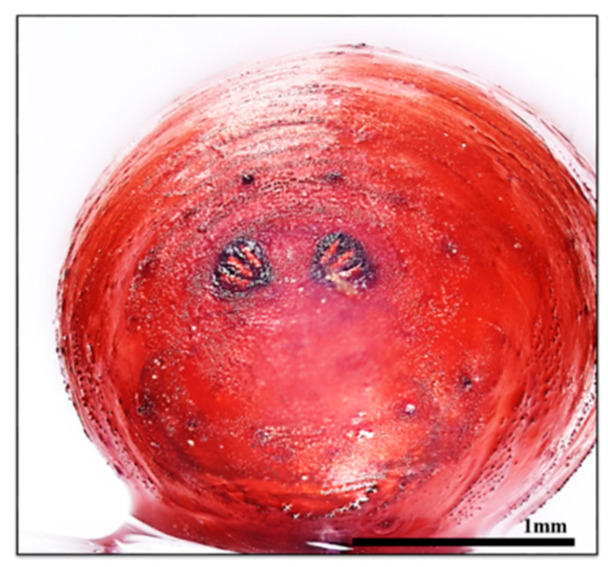
Puparia of *L. sericata* collected in Case 1.

**Figure 6 insects-16-00231-f006:**
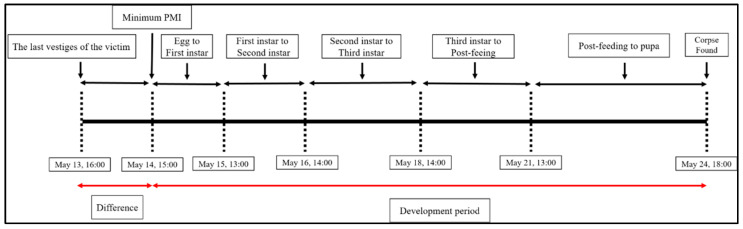
Timeline of Case 1 demonstrating the different life-cycle stages of the insects found on the body, estimated PMI based on the insects, and PMI difference between insect analysis and last known activity.

**Figure 7 insects-16-00231-f007:**
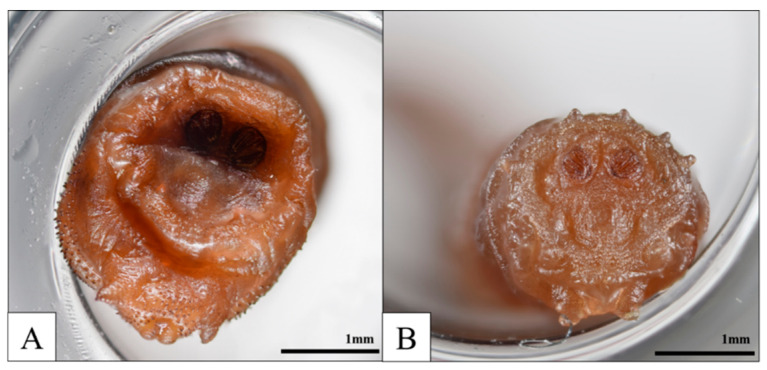
Third-instar larvae collected in Case 2: (**A**) *C. megacephala* and (**B**) *L. sericata*.

**Figure 8 insects-16-00231-f008:**
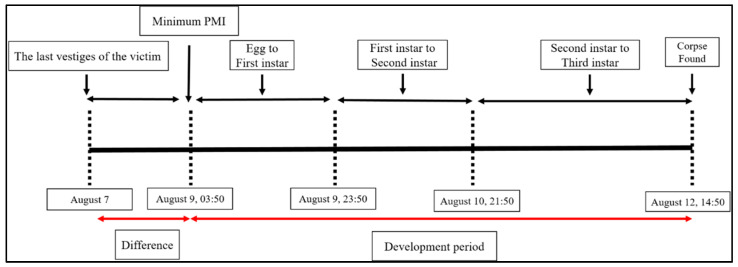
Timeline of Case 2 demonstrating the different life-cycle stages of the insects found on the body, estimated PMI based on the insects, and PMI difference between insect analysis and last known activity.

**Figure 9 insects-16-00231-f009:**
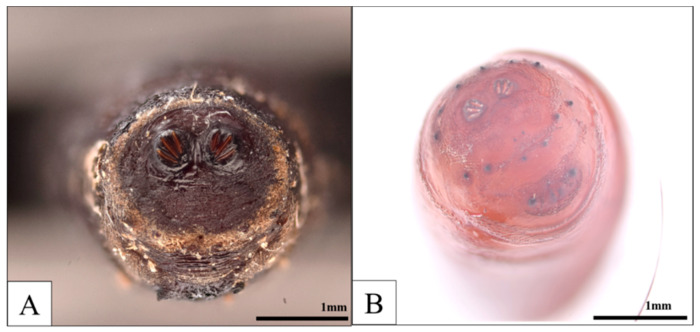
Puparia collected in Case 3: (**A**) *C. megacephala* and (**B**) *L. sericata*.

**Figure 10 insects-16-00231-f010:**
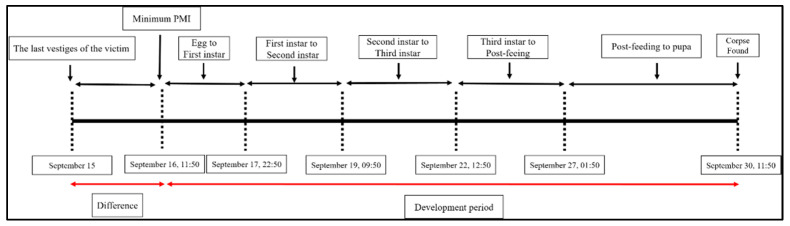
Timeline of Case 3 demonstrating the different life-cycle stages of the insects found on the body, estimated PMI based on the insects, and PMI difference between insect analysis and last known activity.

**Table 1 insects-16-00231-t001:** Primers used for amplification.

Primers	Sequence	Length
F1	CCT TTA GAA TTG CAG TCT AAT GTC A	25
R3	CCA AAG AAT CAA AAT AAA TGT TG	23

**Table 2 insects-16-00231-t002:** Summary of molecular identification results.

Sample ID	Sequence Length (bp)	Species Identification	GenBank Accession No.	Identity (%)
Case 1–1	794	*L. sericata*	JX913756.1	99
Case 1–2	1232	*L. sericata*	JX913756.1	99
Case 1–3	792	*L. sericata*	JX913756.1	99
Case 1–4	1359	*L. sericata*	JX913756.1	99
Case 1–5	786	*L. sericata*	JX913756.1	98
Case 1–6	823	*L. sericata*	JX913756.1	100
Case 1–7	952	*L. sericata*	JX913756.1	99
Case 1–8	966	*L. sericata*	JX913756.1	100
Case 2–1	804	*C. megacephala*	MK075787.1	98
Case 2–2	1200	*C. megacephala*	MK075787.1	99
Case 2–3	794	*L. sericata*	JX913756.1	99
Case 2–4	804	*L. sericata*	JX913756.1	99
Case 2–5	801	*L. sericata*	KT272854.1	99
Case 2–6	1232	*L. sericata*	KT272854.1	99
Case 2–7	797	*L. sericata*	KT272854.1	99
Case 2–8	1181	*L. sericata*	KT272854.1	99
Case 3–1	899	*L. sericata*	JX913756.1	99
Case 3–2	891	*L. sericata*	JX913756.1	99
Case 3–3	832	*L. sericata*	JX913756.1	98
Case 3–4	1336	*L. sericata*	JX913756.1	99
Case 3–5	1228	*C. megacephala*	MK075810.1	99
Case 3–6	1213	*C. megacephala*	MH778895.1	99
Case 3–7	1156	*C. megacephala*	MK075780.1	98
Case 3–8	1184	*C. megacephala*	MK075780.1	99

**Table 3 insects-16-00231-t003:** Summary of the forensic cases studied.

Case Number	Date of Discovery	Entomological Evidence	PMI-min Estimate: Insect Evidence	Estimated Time of Death via Evidence and Autopsy	Time Differences
1	22.05.24. 18h00	*L. sericata* (P)	22.05.14. 15h00	22.05.13. 16h00	23 h
2	22.08.12 14h00	*L. sericata* (L Ⅲ)	22.08.09. 03h00	22.08.07.	2 d
		*C. megacephala* (L Ⅲ)			
3	22.09.30 11h00	*L. sericata* (L Ⅲ, P)	22.09.16. 11h50	22.09.15.	1 d
		*C. megacephala* (L Ⅲ, P)			

## Data Availability

Data are contained within the article.

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
