# Peer review of "Medical–Legal Entomology in Action: Evaluation of Insect-Based Post-Mortem Interval Estimation in South Korean Death Investigations"

_insects, 2025, doi:10.3390/insects16020231_

Round 1
Reviewer 1 Report
Comments and Suggestions for Authors
The article is well written. Introductions can be corrected to include some more broad aspect of conducted research. Methods are well described and correct. Did authors use only molecular identification? If not, appropriate identification keys should be used. How many sequences have been obtained? Did Authors upload them to NCBI database? How good was the match of query sequences with database? Please provide this information to allow for verification. Discussion can be extended. What do you mean by application of machine learning models? I believe more filed works and experiments are necessary to fill gaps in our understanding of carrion succession patterns, not only application of machine learning.
Line 14: insert a space in "(Fabricius)were".
Line 16: explain abbreviation "PMI"
Line 47: autocorrection, should be livor mortis! There is no such thing as liver mortis. Also "livor, algor, and rigor mortis", Latin names are recommended to be in italics.
Line 76 to 78: entire sentence is confusing.
Line 84: why only 70 °C for killing larvae? Recommended temperature of water for killing larvae is app. 90 °C. Please check DOI: 10.1016/j.forsciint.2003.08.010
Line 94: please provide names of primers if universal primers used or sequences if custom primers have been used.
Line 187: Name of author of species description (Meigen or Fabricius) should be used only when species is mentioned for the first time in the manuscript, not every time species name is given.
Line 258: again, correct to livor mortis.
Line 275: I recommend to extend discussion on factors affecting discrepancies between calculated minPMI and last time person was seen alive. For example, discuss factors affecting colonisation:
Matuszewski S., Szafałowicz M., Grzywacz A. 2014. Temperature-dependent appearance of forensically useful flies on carcasses. International Journal of Legal Medicine 128: 1013-1020
Reibe and Madea (2010) doi:10.1016/j.forsciint.2009.11.009
Author Response
Comment 1: The article is well written.
Introductions can be corrected to include some more broad aspect of conducted research.
Response 1: Thank you for your comments. I complement the introduction.
Line 64-67
Validating PMI-min estimation using insect evidence is a critical process, as it not only helps to identify potential errors in the methodology but also provides opportunities for refinement and improvement in the accuracy and reliability of these forensic techniques.
Comment 2: Methods are well described and correct.
Did authors use only molecular identification? If not, appropriate identification keys should be used. How many sequences have been obtained?
Response 2: Thank you for your comments. We used only molecular identification. The information about query sequence was proived in table 2.
Line 277-280
In three cases, a total of two species, L. sericata and C. megacephala, were identified, with L. sericata being present in all three cases (Table 2). These two species are known to predominantly appear in urban areas within the South Korea [11, 19].
Comment 3: Did Authors upload them to NCBI database? How good was the match of query sequences with database? Please provide this information to allow for verification.
Response 3: Thank you for your comments. We will upolad the sequence to NCBI soon. The information about match of query sequence was proived in table 2.
Line 277-280
In three cases, a total of two species, L. sericata and C. megacephala, were identified, with L. sericata being present in all three cases (Table 2). These two species are known to predominantly appear in urban areas within the South Korea [11, 19].
Comment 4: Discussion can be extended. What do you mean by application of machine learning models? I believe more filed works and experiments are necessary to fill gaps in our understanding of carrion succession patterns, not only application of machine learning.
Response 4: Thank you for your comments. The machine learning model I mentioned focuses on factors influencing insect development, such as past temperature conditions at indoor crime scenes.
Comment 5: Line 14: insert a space in "(Fabricius)were"
Response 5: Thank you for your comments. It has been revised.
Comment 6: Line 16: explain abbreviation "PMI"
Response 6: Thank you for your comments. It has been revised.
Comment 7: Line 47: autocorrection, should be livor mortis! There is no such thing as liver mortis. Also "livor, algor, and rigor mortis", Latin names are recommended to be in italics.
Response 7: Thank you for your comments. It has been revised.
Comment 8: Line 76 to 78: entire sentence is confusing.
Response 8: Thank you for your comments. The sentences have been revised, and the modified sentence was moved to the end of section 2.1 to improve the logical flow.
Line 88-91
During DNA barcoding, the most developed samples were prioritized for analysis to estimate the minimum postmortem interval. Additionally, samples with diverse morphological characteristics, such as surface texture and color, were selected to represent a variety of species.
Comment 9: Line 84: why only 70 °C for killing larvae? Recommended temperature of water for killing larvae is app. 90 °C. Please check DOI: 10.1016/j.forsciint.2003.08.010
Response 9: Thank you for your comments. To ensure the morphological features of the maggots are preserved, we followed the methods described in the referenced literature. (https://doi.org/10.1016/j.jflm.2013.03.007). References have been added at the end of the sentence.
Line 84-86
To halt further development, all larvae were immediately immersed in hot water at a temperature of approximately 70 °C for 30 seconds, for preserving the morphological features of the maggots [14].
Comment 10: Line 94: please provide names of primers if universal primers used or sequences if custom primers have been used.
Response 10: Thank you for your comments. I described the sequence about pimer pair for amplification.
Line 100
The designed primer pair were decribed in Table 1.
Comment 11: Line 187: Name of author of species description (Meigen or Fabricius) should be used only when species is mentioned for the first time in the manuscript, not every time species name is given.
Response 11: Thank you for your comments. It has been revised.
Comment 12: Line 258: again, correct to livor mortis.
Response 12: Thank you for your comments. It has been revised.
Comment 13: Line 275: I recommend to extend discussion on factors affecting discrepancies between calculated minPMI and last time person was seen alive. For example, discuss factors affecting colonisation:
Matuszewski S., Szafałowicz M., Grzywacz A. 2014. Temperature-dependent appearance of forensically useful flies on carcasses. International Journal of Legal Medicine 128: 1013-1020
Reibe and Madea (2010) doi:10.1016/j.forsciint.2009.11.009
Response 13: Thank you for your comments. We extend the discussion about fators that affeciting colonization.
Line 290-297
PCI affected several factors like location of the corpse (e.g., indoor, outdoor), ambient temperature, volatile organic compounds (VOCs) [21-24]. According to Reibe & Madea, insect colonization of corpses placed indoors was delayed up to 24 hours compared to those placed outdoors. VOCs emitted from decomposing remains attract carrion insects, and their release is influenced by temperature [25-29]. Although PCI is often overlooked, it is crucial to provide a more accurate PMI estimation. Studies from abroad, such as those by Faris et al. [30], have considered PCI, including exposure, detection, and acceptance phases.
Reviewer 2 Report
Comments and Suggestions for Authors
Notice that the ICZN strongly recommends the year of citation when referencing the species authority [e. g. Chrysomya megacephala (Fabricius, 1754)]. Also, after the first citation, you can subtract the authority from the next ones.
Analyzing the paper as it is, I suggest that you change your approach to a case study, which would be more relevant, and then lead the discussion to the need to use PCI, since you found no statistical difference when comparing the PMI using entomological evidence with that determined by autopsy.
As a final suggestion, the discussion should be improved since you say the PCI is overlooked, but your group didn't use it either. Try not to limit your paper to South Korea, turn it worldly.
Author Response
Comments 1: Notice that the ICZN strongly recommends the year of citation when referencing the species authority [e. g. Chrysomya megacephala (Fabricius, 1754)]. Also, after the first citation, you can subtract the authority from the next ones. |
||
Response 1: Thank you for your comments. It has been revised. |
||
Comments 2: Analyzing the paper as it is, I suggest that you change your approach to a case study, which would be more relevant, and then lead the discussion to the need to use PCI, since you found no statistical difference when comparing the PMI using entomological evidence with that determined by autopsy. |
||
Response 2: Thank you for your comments. It has been revised. Line 282-285 There are no statiscal difference when comparing the PMI using entomological evidence with that determined by autopsy. The slight discrepancies may have arisen because of the differences in the pre-colonization interval (PCI). |
||
Comments 3: As a final suggestion, the discussion should be improved since you say the PCI is overlooked, but your group didn't use it either. Try not to limit your paper to South Korea, turn it worldly. |
||
Response 3: Thank you for your comments. The PCI values for Lucilia sericata were applied based on data derived from the referenced literature. Temperature-dependent appearance of forensically useful flies on carcasses. DOI 10.1007/s00414-013-0921-9 Line 286-289 According Matuszewski et al. [20], minimum PCI value for adult L. sericata in an environment of 14.0-25.1 °C is reported to be 0.2 day. By applying the PCI value to Case 3, which corresponds to the specified temperature range, the PMI-min is calculated to be approximately 20 hours. |

Reviewer 3 Report
Comments and Suggestions for Authors
The manuscript reported three cases in South Korean death investigations in which forensic entomology was used to estimate the minimum postmortem interval. In my opinion, the manuscript has limited value for publication, mainly for the following reasons. 1. There is nothing special about these three cases, and they are all cases that forensic entomology researchers often encounter. Hence, this manuscript does not contribute to the future practice of forensic entomology, nor does it have implications for future research. In addition, the insects reported on corpses are also common species. 2. The introduction of the manuscript is too short and superficial. In addition, the introduction contains some incorrect expressions. For example, insect development patterns were used to estimate the PMImin, while succession patterns were used to estimate the range of PMI. 3. There are only 18 references in the entire manuscript, which is obviously insufficient. This leads to a lack of comparisons to previous studies or cases in the discussion. Considering the above reasons, I don't think the manuscript can be published in Insects at current stage.
Author Response
The manuscript reported three cases in South Korean death investigations in which forensic entomology was used to estimate the minimum postmortem interval. In my opinion, the manuscript has limited value for publication, mainly for the following reasons.
Comment 1: There is nothing special about these three cases, and they are all cases that forensic entomology researchers often encounter. Hence, this manuscript does not contribute to the future practice of forensic entomology, nor does it have implications for future research. In addition, the insects reported on corpses are also common species.
Response 1: Thank you for your comments. I also agree that the species observed in the three cases were common. However, the aim of this study is to analyze the discrepancy between PMI values estimated using insect evidence and the victim's last known activity. Therefore, this manuscript has significant implications for future research in forensic entomology.
Comment 2: The introduction of the manuscript is too short and superficial. In addition, the introduction contains some incorrect expressions. For example, insect development patterns were used to estimate the PMImin, while succession patterns were used to estimate the range of PMI.
Response 2: Thank you for your comments. Incorrect expression was reviesd and the necessity of the study has been included.
Line 56-59
the succession and development of necrophagous insects are utilized to estimate the postmortem interval (PMI) in criminal cases [9]. Succession data indicate the range of PMI whereas development data estimates the PMI-min.
Line 64-67
Validating PMI-min estimation using insect evidence is a critical process, as it not only helps to identify potential errors in the methodology but also provides opportunities for refinement and improvement in the accuracy and reliability of these forensic techniques.
Comment 3: There are only 18 references in the entire manuscript, which is obviously insufficient. This leads to a lack of comparisons to previous studies or cases in the discussion.
Response 3: Thank you for your comments. The maunuscript has been thoroughly revised and refined with careful reference to previous studies and relevant literature.
Round 2
Reviewer 1 Report
Comments and Suggestions for Authors
In Line 280 authors mention "Reibe & Madea", but not such reference is present in reference list.
In reference
Matuszewski, S. Estimating the pre-appearance interval fromtemperature in Creophilus maxillosus L. (Coleoptera: Staphylinidae). J Forensic Sci. 2012, 57, 136–145.
and
VonHoermann, C., Ruther, J., Reibe, S., Madea, B., Ayasse, M. Theimportance of carcass volatiles as attractants for the hide beetle Dermestes maculatus (De Geer). Forensic Sci Int. 2011, 212, 173–179.
2012 and 2011 (year of publication) should be bolded.
Reference:
Shin, S. E., Jang, M. S., Park, J. H., Park, S. H. (2015). A forensic entomology case estimating the minimum postmortem interval using the distribution of fly pupae in fallow ground and maggots with freezing injury. The Korean Journal of Laboratory Medicine, 39(1), 17-21.
does not fit to journal style.
Other changes are correct and manuscript can be accepted after minor correction.
Author Response
Comment 1:
In Line 280 authors mention "Reibe & Madea", but not such reference is present in reference list.
In reference
Matuszewski, S. Estimating the pre-appearance interval fromtemperature in Creophilus maxillosus L. (Coleoptera: Staphylinidae). J Forensic Sci. 2012, 57, 136–145.
and
VonHoermann, C., Ruther, J., Reibe, S., Madea, B., Ayasse, M. Theimportance of carcass volatiles as attractants for the hide beetle Dermestes maculatus (De Geer). Forensic Sci Int. 2011, 212, 173–179.
2012 and 2011 (year of publication) should be bolded.
Reference:
Shin, S. E., Jang, M. S., Park, J. H., Park, S. H. (2015). A forensic entomology case estimating the minimum postmortem interval using the distribution of fly pupae in fallow ground and maggots with freezing injury. The Korean Journal of Laboratory Medicine, 39(1), 17-21.
does not fit to journal style.
Response 1: Thank you for your comments. It has been revised.

Reviewer 2 Report
Comments and Suggestions for Authors
The text in lines 57-59 is confusing and needs to be rewritten.
Author Response
Comment 1: The text in lines 57-59 is confusing and needs to be rewritten.
Response 1: Thank you for your comments. It has been revised.
Line 57~59
Medical-legal entomology, a vital area of forensic science, utilizes the succession and developmental stages of necrophagous insects to enhance the accuracy of criminal investigations [9].
